# Governance of a Blockchain-Enabled IoT Ecosystem: A Variable Geometry Approach

**DOI:** 10.3390/s23229031

**Published:** 2023-11-07

**Authors:** Ikram Ullah, Paul J. M. Havinga

**Affiliations:** Pervasive Systems Group, Department of Computer Science, University of Twente Enschede, 7522 NB Enschede, The Netherlands; p.j.m.havinga@utwente.nl

**Keywords:** Internet of Things, blockchain, governance, European Union policy, EU policy, policy makers, International Organization for Standardization, ISO

## Abstract

The proliferation of Internet of Things (IoT) applications is rapidly expanding, generating increased interest in the incorporation of blockchain technology within the IoT ecosystem. IoT applications enhance the efficiency of our daily lives, and when blockchain is integrated into the IoT ecosystem (commonly referred to as a blockchain-IoT system), it introduces crucial elements, like security, transparency, trust, and privacy, into IoT applications. Notably, potential domains where blockchain can empower IoT applications include smart logistics, smart health, and smart cities. However, a significant obstacle hindering the widespread adoption of blockchain-IoT systems in mainstream applications is the absence of a dedicated governance framework. In the absence of proper regulations and due to the inherently cryptic nature of blockchain technology, it can be exploited for nefarious purposes, such as ransomware, money laundering, fraud, and more. Furthermore, both blockchain and the IoT are relatively new technologies, and the absence of well-defined governance structures can erode confidence in their use. Consequently, to fully harness the potential of integrating blockchain-IoT systems and ensure responsible utilization, governance plays a pivotal role. The implementation of appropriate regulations and standardization is imperative to leverage the innovative features of blockchain-IoT systems and prevent misuse for malicious activities. This research focuses on elucidating the significance of blockchain within governance mechanisms, explores governance tailored to blockchain, and proposes a robust governance framework for the blockchain-enabled IoT ecosystem. Additionally, the practical application of our governance framework is showcased through a case study in the realm of smart logistics. We anticipate that our proposed governance framework will not only facilitate but also promote the integration of blockchain and the IoT in various application domains, fostering a more secure and trustworthy IoT landscape.

## 1. Introduction

In this paper, we introduce a governance mechanism for blockchain-enabled IoT ecosystems, utilizing a variable geometry approach. Establishing an effective governance mechanism is of utmost importance in the context of blockchain-enabled IoT ecosystems to coordinate the planning, execution, and monitoring of the associated infrastructure. In the subsequent sections, we will commence by introducing IoT applications, proceed to elucidate the integration of blockchain and the IoT, and culminate by delving into the pivotal role of governance in the realm of blockchain-enabled IoT systems.

### 1.1. IoT Applications

The integration of the IoT into our daily lives offers a range of innovative applications. Technological advancements have profoundly transformed our way of life, enhancing communication, simplifying daily routines, and fundamentally reshaping various industries, as shown in Figure 1. IoT technology is currently in its early stages, and numerous new innovations are rapidly emerging within the IoT landscape. The quantity of interconnected devices is expanding swiftly, with projections suggesting that by 2025, the total number of connected devices could potentially soar to 100 billion [1,2]. The proliferation of IoT devices could potentially result in exponential data growth [3]. Data are collected, stored, and processed to support a variety of IoT services. The IoT infrastructure comprises diverse components, including sensors, actuators, RFID, Wireless Sensor Networks (WSNs), cloud systems, and big data solutions. Cloud computing has the potential to significantly enhance the success of the IoT due to its numerous advantages, such as easy implementation, cost-effectiveness, efficiency, and ability to handle large datasets. Cloud computing proves particularly beneficial when the IoT generates substantial data and when sensors are distributed across geographic locations [4]. The integration of the IoT, cloud, and big data mechanisms serves as the foundation for numerous contemporary innovations [3].

Besides its wide-ranging applications, the IoT also faces several challenges [5]. IoT solutions are still in the nascent stages of development, and in many instances, these solutions are intricate. This complexity arises for several reasons, including issues related to interoperability, communication, data volume, real-time data analysis, demanding development cycles, and standardization [3,6,7]. The predominant concerns related to the IoT encompass privacy, security, standards, governance, and ethics [3,8,9]. Furthermore, additional challenges in the realm of the IoT are detailed in [10,11,12]. Beyond the IoT, numerous challenges are linked with cloud computing, including concerns about confidentiality, trust, privacy, integrity, and the unauthorized storage and processing of personal data. Advancements in technology have made it easier for criminals to commit various forms of cybercrime. Criminals have escalated their tactics, automated their attack methods, refined mechanisms for extracting ransom payments in cryptocurrencies, and modernized their business models, as evidenced by the use of different languages (such as German in the case of the “WannaCry” ransomware targeting German rail systems).

### 1.2. Blockchain and IoT Inclusion

Blockchain is a distributed peer-to-peer network, where the nodes run blockchain protocols to validate transactions. Transactions are transparently stored in distributed ledgers, and each node has a copy of the ledger. Blockchain allows the participants of a network to achieve consensus over the shared ledger without the need for any central party or human interactions [13,14]. Emerging technologies like Artificial Intelligence (AI), the IoT, blockchain, and robotics are pivotal in driving the Industry 4.0 revolution. Blockchain, in particular, are a groundbreaking and innovative technology with diverse applications across various sectors. While there is a multitude of blockchain applications, some notable examples include e-voting, cryptocurrency, healthcare [15], automotive [16], sustainability [17], supply chain [18], insurance [19], and procurement services [20,21,22]. This technology has garnered substantial interest from investors, start-ups, venture capitalists, and industries, with over USD 3 billion invested in blockchain start-ups in 2019 [23,24]. Beyond the business realm, government organizations, policymakers, scholars, and regulators are also exploring the potential of blockchain technology [25,26,27,28]. To address the challenges faced by the IoT, integration with blockchain technology presents a promising solution. Blockchain can address IoT security, transparency, trust, and privacy concerns. Nevertheless, the integration of these novel technologies introduces its own set of challenges, with one of the primary concerns being the lack of a dedicated governance structure. In this paper, we put forth an efficient and appropriate governance framework for a blockchain-enabled IoT ecosystem.

### 1.3. Governance

Governance is “a system of decision rights and accountabilities for information-related processes, executed according to agreed-upon models which describe who can take what actions with what information, and when, under what circumstances, using what methods” [29]. In the context of the IoT, security, privacy, and governance are intertwined; governance is essential to establish trust, security, and privacy [3,9]. Governance aims to define roles, policies, and responsibilities to achieve objectives such as interoperability, liability, security, privacy, and trust. Although security, privacy, and governance are closely related, they are not identical. Security mechanisms ensure data protection from malicious purposes, whereas privacy mechanisms dictate how to collect, process, and store users’ data. A governance framework encompasses policies and processes related to various controls (e.g., security, privacy, business practices, and organizational controls) in place. Through robust governance mechanisms, organizations can implement policies, processes, accountabilities, and roles and make informed decisions to efficiently manage corporate resources. Well-designed and implemented governance mechanisms are crucial for addressing user and stakeholder concerns [30]. The World Bank’s governance consists of mechanisms and institutions which “includes the process by which governments (governing body) are selected, monitored and replaced; the capacity of the government to effectively formulate and implement sound policies; and the respect of citizens and the state for the institutions that govern economic and social interactions among them” [31]. With the evolution of emerging technologies, conventional governance mechanisms, which constitute people, processes, and technologies, are evolving [32,33] into more rigorous frameworks. The introduction of the General Data Protection Regulation (GDPR) [34] in Europe has made governance even more indispensable [7], with the potential to facilitate IoT development [4] and adoption. With the advent of distributed ledger technologies such as blockchain and the inherently distributed nature of the IoT architecture, the concept of governance has gained significance. However, it has also become more challenging, as traditional centralized governance mechanisms are no longer applicable [35]. While the technical aspects of the IoT are widely discussed, adequate legal frameworks are yet to be established [36]. The European Commission has encouraged experts to explore the potential features of governance [36], becoming the first international organization with the goal of establishing a governance framework [8].

Figure 2 illustrates the core principles essential for fostering and guaranteeing an inclusive governance framework. These principles encompass democracy and ethics, confidence, collaboration, innovation, well-being, and economic prosperity. The ultimate aim of governance is to realize these pivotal aspects in the context of the modern world. Governance policies should be anchored in the foundational pillars of democracy and the principles of ethics. Transparency and disclosure are regarded as the core characteristics of a governance framework [37]. Embracing change, particularly when adopting new technologies, can be challenging due to the inherent lack of confidence and concerns about data security among users. Achieving consensus is often hampered by competition among stakeholders or a lack of trust [38]. Confidence in new technologies is crucial, as it can lead to success and ultimately, the large-scale adoption of technology and, consequently, collaboration among many partners and stakeholders. In new technologies where personal data are involved, there are many consumer concerns such as “fear of unknown” [30]. Governance plays a pivotal role in facilitating the future development [4], adoption, and endorsement of novel technologies. An effective governance framework must encompass policies that address the concerns of users and stakeholders, ensuring their confidence is maintained through the implementation of mechanisms and policies that guarantee confidentiality, integrity, and availability. User confidence in new technologies is instrumental in driving mass adoption and catalyzing innovation.

Governance serves as a catalyst for collaboration among diverse partners and industries. While individual technologies can bring benefits, the integration of various technologies can yield the most exquisite innovations, enhancing efficiency across various application domains. Partners can collaborate to construct a shared technological infrastructure, with each partner contributing dedicated knowledge, expertise, and resources, ultimately generating value. Governance policies play a pivotal role in assuring that these applications not only enhance the quality of life [30] but also reduce environmental impacts (e.g., through recycling physical objects) and enhance environmental sustainability by extending the lifetime of the technology.

Governance can be classified into various categories. As mentioned in [38,39,40], governance frameworks are categorized into three primary groups: markets (focused on individual choice), hierarchies (centered around formal organizational structures), and networks (emphasizing consensus and blockchain technology). These categories can be further subdivided into various types of frameworks, which are extensively discussed in the literature and widely applied in practice. Examples include IT governance [41], IoT governance [42], cloud governance [4], social-political governance [43], and numerous other generic and industry-specific frameworks. For instance, the National Institute of Standards and Technology (NIST) [44] has established comprehensive standards for cybersecurity. For instance, NIST SP 800-53 (Security and Privacy Controls for Federal Information Systems and Organizations) [44] describes security and privacy controls for federal information systems and organizations. The General Data Protection Regulation (GDPR) [34] was introduced by the EU to protect citizens’ privacy and rights regarding their data. This standard is applied to European Union organizations and businesses, particularly for EU residents. Fines can be significant if GDPR policies are violated. Marriott was fined USD 123 million for a 2018 data breach [45]. ISO 27000 [46] is an internationally recognized family of standards. For instance, ISO/IEC 30141 [47] is aimed at standardizing the IoT reference architecture to ensure that the IoT ecosystem is “seamless, safer, far more resilient”. ISO/IEC 27001 [48] applies to cyber security. Other related standards are ISO 270018 [49] and ISO 270017 [50]. ISO 27000 [46] ensures information assets security. These various standards and frameworks can be simultaneously implemented in practice. Furthermore, every country has a dedicated body for standards and compliance. For instance, in the Netherlands, The Royal Netherlands Standardization Institute (NEN) [51] is responsible for standardization.

Each of these governance frameworks serves distinct purposes and differs significantly from the others. The IoT stands apart from the traditional Internet in several aspects, such as implementation, maintenance, development, ethical considerations, privacy, security [30], and the incorporation of emerging technologies like blockchain. The characteristics and requirements of the IoT extend beyond the scope of conventional Internet governance [52] and other governance frameworks. Therefore, the establishment of a contemporary blockchain-enabled IoT governance framework becomes essential. Nonetheless, the knowledge derived from Internet governance [8] and other governance frameworks remains invaluable and can play a pivotal role in the development of a comprehensive blockchain-enabled IoT governance system [30]. For example, when considering Internet governance, which predates IoT governance, close collaboration with Internet governance bodies becomes essential [30]. The integration of existing governance mechanisms can effectively complement the governance of blockchain-enabled IoT systems. As illustrated in Figure 3, various standards and governance frameworks are incorporated to ensure a comprehensive and expansive approach to blockchain-enabled IoT governance.

As per the EU IoT task force, the IoT differs from general Internet governance and thus further research, separate rules, and regulations are essential [53]. It is evident that there is a notable deficiency in comprehensive guidance pertaining to the governance of the IoT [32].

As previously indicated, there is no dedicated legal IoT governance framework yet [3]. The lack of a matured governance is one of the many IoT challenges [3]. For instance, the CASAGRAS project recommends a dedicated governance framework for the IoT on a global and regional level [53]. The field of IoT governance has seen the least research and progress since the IoT revolution in the last decade. Developing strategic IoT governance mechanisms has unparalleled effects on the overall sustainability of the IoT with regard to environments and finances. Copie et al. [4] discussed various use cases to highlight the importance of governance. For instance, IoT governance allows us to manage IoT processes to add or revoke a device, aggregate data from multiple sources, and establish policies regarding roles, security, privacy, trust, and data storage [4]. One main reason behind the lack of governance mechanisms is that developing and managing an international legal framework is not so straightforward, as different countries and regions have different legislation. With the integration of blockchain and the IoT, the development of an adequate governance framework becomes highly desirable.

As previously noted, the comprehensive design and implementation mechanisms put forth for governance are essential for addressing users’ concerns [30] and fostering innovation in the future.

Hence, we recognize the need for a more dynamic and vibrant governance mechanism that includes roles and policies both at the societal level and information level and utilizes blockchain characteristics (i.e., immutability, traceability, decentralization) for the governance aspiration. Blockchain can demonstrate governance features that are more cost-efficient, deliver greater trust [37], and allow for the development of decentralized governance, which is far more efficient and automated. However, with the number of blockchain frameworks increasing significantly, there is no concrete blockchain governance mechanism. Blockchain governance is ”the means of achieving the direction, control, and coordination of stakeholders within the context of a given blockchain project to which they jointly contribute” [54]. By deploying blockchain technology in various applications, we can achieve most of the principles mentioned in Figure 2, since blockchain encourages collaboration and potentially presents distinctive governance features [23].

The remainder of this paper is structured as follows. In Section 2, we delve into the IoT architecture and reference models. Section 3 offers a brief introduction to the technical aspects of blockchain. Section 4 provides an overview of related works, while Section 5 introduces our proposed governance framework. Lastly, Section 6 presents an evaluation of the proposed framework.

## 2. IoT Architectures and Reference Model

The IoT includes many different types of devices, protocols, and architectures. Therefore, there is no unified IoT reference architecture, since a single reference architecture might technically not suffice; thus, multiple reference architectures can simultaneously exist [8]. An internationally standardized reference architecture guarantees that connected systems are “seamless, safer and far more resilient” [55]. Various layered architectures [47,56] have been proposed, for instance, three-layered [57] and five-layered architectures [57,58]. The three-layered architecture is fundamental, although inadequate due to the continual innovation in the field of the IoT [59], and it cannot fulfill all the requirements of the IoT for diverse application domains. The three-layered architecture consists of perception, network, and application layers, whereas the five-layered architecture consists of perception, network, middleware, application, and business layers, as demonstrated in Figure 4. The six-layered architecture is the new optimized IoT reference architecture. Blockchain technology is embedded in a six-layered IoT architecture. An illustration of a six-layered IoT reference architecture is shown in Figure 4. In IoT implementations, it is important to realize the IoT architecture in order to understand the roles and features of every layer in the architecture, enable compatibility, and, consequently, IoT deployment. Below, we describe the six-layered architecture, along with the security mechanisms and threats associated with each of these six layers. However, the impacts of these threats can vary, as they depend on the specific IoT application domain.

### 2.1. Perception Layer

The perception layer is the physical layer and mainly consists of sensors, RFID tags, and actuators. Security mechanisms required for this layer include data protection, physical security, lightweight encryption, and key management [59,60]. As most of these devices are resource-constrained, a simple security mechanism can be applied here; therefore, IoT devices are mostly vulnerable. The most common threats to the perception layer are eavesdropping, node capture, replay attacks, timing attacks [59], Sybil attacks, and sniffing attacks.

#### Roles and Objectives

This layer is responsible for device identification and data collection. The data collected are forwarded to the network layer [57].

### 2.2. Network Layer

This layer consists of wired and wireless communication protocols such as ZigBee, Bluetooth, and WiFi. Frequent security threats to this layer are denial of service (DoS), man-in-the-middle, and storage attack [59]. Security mechanisms required for this layer are identity authentication, encryption, communication security, and trust mechanisms.

#### Roles and Objectives

The role of this layer is to transmit sensor data to data processing entities using communication protocols.

### 2.3. Blockchain Layer

The blockchain layer is a new bridging layer between the network layer and the middleware layer for a blockchain-enabled IoT ecosystem. The blockchain layer is also vulnerable to various threats such as double-spending, identity theft, illegal activities, denial of service (DoS) attacks, reply attacks, and system hacking [61,62,63].

#### Roles and Objectives

Blockchain characteristics, such as immutability, consensus, traceability, decentralization, and automated execution of rules, are utilized in this layer for transparency, security, privacy, and trust purposes.

### 2.4. Middleware Layer

Numerous IoT devices are deployed to implement diverse IoT services. IoT devices communicate and exchange data to deliver these services. This layer consists of ubiquitous computing, integration, data analytics, service management, and databases. Attacks possible at this layer include DoS and malware [59] attacks. Security mechanisms required for this layer are authentication, encryption, communication security, and trust mechanisms.

#### Roles and Objectives

The purpose of this layer is service management and database management [56]. The data received from the network layer are further processed to extract useful information and execute service-oriented decisions.

### 2.5. Application Layer

This layer consists of application programming interfaces (APIs), interfaces, and data representations. Attacks possible at this layer include cross-site scripting, malicious code attacks, sniffing attacks, phishing attacks, and sensitive data leakage [59]. Common security mechanisms required to secure the application layer are access control, authentication, key management, encryption, API protection, and trust mechanisms.

#### Roles and Objectives

The purpose of this layer is to provide application management for IoT applications such as smart logistics and smart agriculture.

### 2.6. Business Layer

This layer consists of business models, business requirements, and visualizations. Attacks possible at this layer include business logic attacks and zero-day attacks [59]. Security mechanisms required for this layer are secure business logic programming and secure data exchange.

#### Roles and Objectives

This layer is responsible for efficiently managing the IoT system, executing business-related decisions in order to attain business profits, ensuring users’ privacy, and planning future strategies.

## 3. Blockchain

Blockchain, which is a peer-to-peer network, is an alternative to centralized network architectures. Existing centralized networks rely heavily on intermediaries, and these intermediaries pose significant risks such as data tampering. Interference by third parties or intermediaries can lead to various potential breaches [64,65]. Therefore, a decentralized blockchain technology, without trusted intermediaries, is required for secure and trustless networks. Blockchain comprises “blocks” consisting of transactions, and these “blocks” are cryptographically linked to form a “chain”. A block’s data field mainly consists of the block number, hash of the block data, size of the block, transactions, time stamp, hash of the previous block, and a nonce. Peers verify blocks using cryptographic hashes [64]. Genesis (initial state of the chain) is the first block of the blockchain network. Other blocks are added based on the underlying consensus algorithm. Blocks are chained together by referencing a previous block’s hash. Since blocks reference previous blocks, if a block is changed, the hash value will change and thus all the succeeding hashes will change [66]. The way blockchain works totally depends on the application domain or requirements; there are no standardized specifications to design and employ blockchain. Blockchain developers and architectures are free to select any combination of algorithms. Figure 5 shows how blockchain works:Public key cryptography can be used to communicate with the blockchain. Users are identified by their public keys, and their private keys can be used to digitally sign transactions.In a peer-to-peer network, a peer generates digitally signed transactions that contain the transfer of funds.The signed transactions are broadcast over the peer-to-peer network.Neighboring peers validate the transactions and spread the transactions across the entire network.Miners form blocks of the validated transactions.Blocks are broadcast over the network.The nodes verify and validate the blocks, and validated blocks are added to the ledger.Eventually, transactions are executed.

Blockchain enables fast transactions since there are no centralized intermediaries. Furthermore, blockchain allows for the storage of data in a transparent, verifiable, and immutable manner in the distributed ledger. Every node in the blockchain network keeps a copy of the ledger. Various blockchain permission models have been proposed. In a permissionless model, every peer in the network can participate and add a new block to the ledger without requiring permission to join the network. In a permissionless model, no trust is required among the peers in order to communicate and execute transactions. In a permissionless network, peers identify other peers through their public addresses. However, malicious users might attempt to manipulate the ledgers or add erroneous blocks. In order to circumvent such malicious behavior, blockchain uses consensus algorithms, where peers are required to demonstrate certain capabilities or resources [66]. Through consensus algorithms, the nodes verify and validate the transactions. In a permissioned blockchain network, only authenticated peers are allowed to participate and add new blocks; only whitelisted peers can read and write to the ledger [66]. Since only trusted peers can execute transactions in the permissioned model, consensus mechanisms are typically faster [66]. In a permisssioned blockchain network, to some extent, there is trust among the participating peers [66].

Consensus “is the process in which a majority (or in some cases all) of network validators come to an agreement on the state of a ledger. It is a set of rules and procedures that allows maintaining coherent set of facts between multiple participating nodes” [67]. In blockchain technology, consensus protocols are used to obtain a consensus on transactions or ledgers. In order to reach a consensus, peers are required to agree on transactions; otherwise, forks are formed. In forks, each peer has a distinct copy of the ledger. In a permissionless network, many peers compete to add the next block to the chain. The peer that adds the block is rewarded with transaction fees. On average, Bitcoin consensus requires 10 min before adding a block to the ledger [64]. Once blocks are added to the ledger, it can not be modified since blockchain records are immutable. When two peers solve the block at the same time, forks are formed and peers have different ledgers. To solve this conflict, the longest chain is regarded as the valid chain and the peers embrace it [66]. Furthermore, technologies are regularly updated to improve performance or add new features. The updating of a blockchain protocol is also called a fork. There are two types of forks: soft and hard forks. Soft forks are changes in blockchain protocols that are backward compatible, whereas hard forks are changes that are not backward compatible. As mentioned earlier, forks are normally formed when multiple miners solve the block at the same time or there are conflicts in the ledger. However, there are various mechanisms for resolving such forks.

Some of the well-known consensus algorithms are the proof-of-work (PoW), proof-of-stake (PoS), and practical byzantine fault-tolerance (PBFT) algorithms. These algorithms work significantly differently from each other. Each of these algorithms has both pros and cons. In the proof-of-work (PoW) consensus mechanism, nodes solve computationally complex puzzles in order to add the next block. To solve the puzzle, miners generate hash values that must meet certain requirements. Various hashing algorithms are used in mining [68], such as SHA-256 [69], scrypt [70], and Blake-256 [71]. Bitcoin uses the PoW consensus mechanism [66]. PoW is known to be computationally the most intensive consensus mechanism. Miners perform computationally intensive PoW operations in order to add the next block to the chain. All other nodes can easily verify that the computations performed are correct and then add the block to the chain. Miners are rewarded for their computations. In private blockchain networks, a computationally intensive PoW mechanism is not necessary since there are fewer chances of Sybil attacks [68]. PoW can prevent blockchain networks, to some extent, from denial-of-service attacks [66]. PoS is another consensus mechanism where a block is added to the chain based on the balance (stake) of the peer. Peers with high balances have more chances to add the next block. The advantage of PoS is that it is not computationally intensive. It can be used in permissionless blockchain networks. It is implemented in Ethereum Casper and Krypton [66]. The main disadvantage of this mechanism is that it is vulnerable to a 51% attack, and peers with high stakes can control the network [66]. In the PBFT consensus mechanism, it is assumed that consensus can be reached as long as n=3f+1 correctly working nodes are present in the network, where *f* represents faulty nodes and fewer than 1/3 of the nodes are faulty [72]. And, 2f+1 network nodes are required for consensus on the block. PBFT is an energy-efficient consensus algorithm and is suitable for private (permissioned) blockchain networks. PBFT is implemented in Hyperledger Fabric [73]. The limitation of PBFT is that it is not suitable for large-scale networks. To improve the performance of PBFT, various variations have been proposed. Other consensus mechanisms include the delegate-proof-of-stake (DPoS) [74], proof-of-elapsed-time (PoET) [75], lease-proof-of-stake (LPoS) [76], proof-of-capacity (PoC) [76], and proof-of-interaction (PoI) [76] mechanisms.

Two types of record-keeping models are popular in today’s blockchain networks. The first method is the unspent transaction output (UTXO) model [77] and the second method is the account-based model [77]. The UTXO model is employed by Bitcoin, whereas Ethereum uses the account-based model [77]. A blockchain that supports the UTXO model is uniquely suited for the transfer and tracking of digital tokenized assets, whereas a blockchain that supports the account-based model is aimed at running arbitrary logic and establishing verifiable multi-step processes (smart contracts) [68]. Over the course of time, various applications based on blockchain have been developed. Bitcoin is a standard and well-known application of blockchain. The motive behind blockchain technology extends far beyond cryptocurrency applications [78]. Legislators in various US states are using blockchain for different purposes such as secure records storage and smart contracts [79,80,81]. Track-and-trace mechanisms are very important in IoT applications, particularly in smart logistics. Blockchain technology can provide a trusted infrastructure for tracking and tracing both physical objects and information [78]. Other characteristics of blockchain applications are the immutability of information, transparent sharing of information, and automated business processes [78]. Many different types of blockchain frameworks have been developed for various purposes [25], such as Bitcoin [82], Ethereum [83], Ripple [84], Hyperledger [73], BigchainDB [85], Corda [86], Quorum [87], Tezos [88], Multichain [89], Hashgraph [90], IOTA [91], and R3 [92]. The various classifications of blockchain are presented in [38,93].

### 3.1. The Mechanics of Smart Contracts

A smart contract is a popular application of blockchain technology since it can be implemented easily and efficiently [64] and has a wide range of uses. A smart contract is “a computerized transaction protocol that executes the terms of a contract. The general objectives of smart contract design are to satisfy common contractual conditions (such as payment terms, liens, confidentiality and even enforcement), minimize exceptions both malicious and accidental, and minimize the need for trusted intermediaries” [94]. A smart contract comprises a set of rules and policies that are executed automatically on the blockchain. These rules and policies are programmed into the blockchain. The smart contract enforces immutability and automation through cryptographic mechanisms. When certain predefined conditions are met, smart contracts execute tasks automatically without the need for third-party interference. Therefore, intermediaries or centralized third parties are not required to enforce contracts. Any computational logic or functionality implemented on the blockchain can be regarded as part of the smart contract, which is otherwise manually enforced in traditional contracts. Peers in the blockchain network execute the smart contract, all the peers in the network agree on the results, and the results are recorded on the blockchain. In a permissionless blockchain network, the peer pays a fee for executing a transaction. Smart contracts can be used in various applications such as financial [66], random number generation [95], supply chains, legal, intellectual property, and insurance. Smart contracts have many unique characteristics; they are tamper-evident and tamper-resistant [66], which demonstrates their transparency characteristics. Smart contracts yield many advantages since manual executions are time-consuming and error-prone [64], and they can save up to USD 4 billion in costs spent on errors and manual efforts [96]. Smart contracts are deterministic; a certain input must always generate the same output [66].

Various implementations of smart contracts have been introduced, such as Ethereum smart contracts and Hyperledger Fabric chain code. Ethereum is considered an extension of Bitcoin, as it supports a wide range of applications [64] such as the Ethereum smart contract. As shown in Figure 6, the Ethereum Virtual Machine (EVM), which is a part of Ethereum, is a computation engine, where the smart contract code is executed and handles the deployment of smart contracts. Ethereum is a transaction-based state machine [97]. Miners or account holders initiate transaction execution in order to perform a task. When a smart contract is invoked, EVM changes the Ethereum state as per the Ethereum protocol. The account that requests the transaction pays transaction fees in Ether (Ethereum cryptocurrency). The miner is awarded the fees after the successful execution of the transaction. Ethereum smart contracts are written in a high-level programming language such as Solidity. Solidity is compiled into the EVM bytecode instruction. The EVM bytecode is deployed on the Ethereum blockchain network. The EVM provides a large set of functionalities, such as arithmetic, stack, system, logic, block, and environment operations [97].

### 3.2. Role of Blockchain in Governance

Blockchain is not only beneficial for upholding security, privacy, and trust but it can also be used as a governance mechanism. Contracts are inscribed to align the interests of organizations working together. Contracts are mainly enforced among multiple parties in order to collaborate and cooperate. Contracts aim to legally enforce rights and obligations, and each party is obliged to adhere to the contract conditions. Inter-organizational collaboration is challenging for various reasons such as mistrust among partners [38] or a large number of intermediaries, and overall, the processes are error-prone [79]. However, inter-organizational collaboration is highly important and advantageous [38]. Technologies are adopted to improve transparency and verifiability in collaboration among partners [79]. Exchanging data among many collaborating partners is quite challenging for various reasons, such as data ownership, business secrets, and privacy consequences. Accountability, predictability, and a common understanding are the three conditions required in order to collaborate [23,98]. Governance plays an important role in achieving these three requirements, and therefore, organizations defer to governance mechanisms [23]. The rules and regulations of the contract can be self-enforcing, manual, or enforced legally (national laws). In traditional collaboration, tasks are executed manually, and manual execution is not only error-prone but also time-consuming, potentially leading to conflicts. Blockchain can play an important role in enforcing the contract in a decentralized and transparent manner. For instance, in smart contract scenarios, the rules of the contract are automatically enforced through blockchain technology (protocols and code-based rules) [23]. Blockchain allows for the verification and validation of policies and rules agreed upon among the partners in real time and in a decentralized manner. Furthermore, the immutability, transparency, and traceability [18] characteristics of blockchain can be used to establish intra- and inter-organizational collaboration [23,99]. Other novel characteristics of blockchain include decentralized consensus and machine-based automation [23], which can facilitate fast and reliable collaborations [23], boost confidence in the fairness of data evaluation, and make it quite easy to perform audits in cases of fraud or error. Through decentralized consensus, data integrity can be achieved, and it becomes difficult to tamper with the data [23]. The consensus and smart contract characteristics of blockchain are very beneficial for automatically executing the agreed-upon obligations in a collaborative context involving multiple parties [79]. Thus, blockchain can potentially resolve the existing inefficiencies in multi-party coordination [79].

## 4. Related Works

Various analyses and sentiments can be found in the literature regarding blockchain-IoT integration and governance. Since governance by blockchain and governance for blockchain are two different terms, in this section, we present the related works concerning IoT governance, the role of blockchain as a governance mechanism, and governance for blockchain.

### 4.1. IoT Governance

Numerous governance frameworks have been proposed in the literature. Copie et al. [4] presented IoT governance aspects that are closely related to cloud governance, with the main focus on security, privacy, and standards. The proposed IoT governance is a multi-agent governance architecture with defined roles for agents. The agents include vendor, deployment, proxy, audit, monitoring, aggregation, user interface, software Thing, Thing management, security management, audit management, and governance management. Salazar et al. [42] proposed a generic governance model for IoT solutions, aiming to improve the implementation and management of IoT applications. They described the development strategy, skills, roles, standards, processes, and policies to be incorporated into IoT governance. The processes and policies module includes various principles such as technical management, the complexity of the IoT solution, the device vendor, portfolio management, and operational management. Ruithe et al. [3] proposed a generic data governance framework for the IoT-cloud domain. The authors identified key concerns related to IoT-cloud convergence. The IoT-cloud governance roles (for monitoring security), responsibilities (for providing updates and patches), and policies (for security, privacy, integration, etc.) in relation to security and privacy were presented in the form of a framework. Kazmi et al. [7] proposed a smart governance framework in order to manage heterogeneous IoT devices and enable interoperability across various domains in smart cities. In their proposed mechanism, IoT data and services are integrated from heterogeneous IoT networks and then monitored and governed from a central point. The centralized governance layer provides various services such as ensuring the meeting of business requirements and enforcing security policies. An international framework for IoT governance was proposed in [30]. In this framework, various roles and purposes, which are deemed necessary for implementing international governance, were proposed. Purpose determined the scope and definitions of terms that might be used. Several roles were discussed as potential IoT governance stakeholders such as the government, the private sector, and civil society. Dasgupta et al. [32] proposed a conceptual framework for data governance in an IoT-enabled ecosystem. The authors enhanced the 4I (identify, insulate, inspect, improve) framework in order to emphasize the necessity of vigorous data governance in the IoT-enabled ecosystem. They proposed a Design Science Research (DSR) mechanism to evaluate and extend the 4I framework. Moreover, several existing IoT frameworks, such as Microsoft Azure [100], IBM BlueMix [101], Xively [102], IoTivity [103], and ThingSquare [104], have adopted IoT governance to some extent, including governance for encryption, the cloud, and the device itself.

### 4.2. Role of Blockchain as Governance Mechanism

Lumineau et al. [23] researched the role of blockchain as a governance mechanism within an organization and across organizations. Based on their analysis, blockchain can potentially play an important role in encouraging well-defined cooperation and coordination. They argued that blockchain has the capabilities of efficient governance mechanisms and is distinct from contractual and relational governance mechanisms. Moreover, many real-world examples were mentioned to demonstrate that blockchain can facilitate coordination among various organizations. Various arguments were presented by the authors to highlight applications of blockchain and how blockchain can ultimately be used as a governance mechanism due to its decentralization, automation, and immutability characteristics. The authors of [78] mentioned various characteristics of blockchain to support the role of blockchain as a governance mechanism. The characteristics mentioned were immutability, track and trace, elimination of centralized or third parties, peer-to-peer transactions, automation of processes, and transparent consensus. These characteristics can be beneficial in many application domains such as e-voting, corporate governance, finance, and smart logistics. Elst and Lafarre [79] proposed a governance framework based on a permissioned blockchain to solve various stakeholders’ problems regarding third-party systems of shareholder engagement. These problems were divided into two main categories: (1) lack of transparency and trust between shareholders and (2) information problems and inequalities among shareholders. Based on their discussion, blockchain can be used by shareholders to perform voting, enforce majority requirements for certain decisions, and implement access rights for shareholders without any need for intermediaries. Falco et al. [37] studied the potential impact of blockchain on corporate governance, specifically on the board of directors and institutional investors. A survey was conducted that included 47 respondents from various countries. Based on the respondents’ interview results, blockchain could impact areas such as ownership, voting, turnout rate, market liquidity, and transparency. Qi et al. [105] discussed applications of blockchain in e-governance to enhance service quality and efficiency. Last but not least, the transparency and traceability features of blockchain allow us to more effectively embrace the Shareholder Rights Directive (EU Directive 2017/828) regulations [37]. Furthermore, blockchain can address certain provisions from Shareholder Rights Directive II [106].

### 4.3. Governance for Blockchain

Pelt et al. [25] proposed a conceptual blockchain governance framework. The framework consists of six dimensions and three layers. The six blockchain governance dimensions are (1) formation and context, which illustrates the purpose of the BC; (2) roles, which describe the responsibilities and accountabilities of the participants; (3) incentive, which describes the motives for the assigned roles; (4) membership, which illustrates the mechanism used to manage participation and membership; (5) communication, which illustrates the mechanism used to communicate with the stakeholders; and (6) decision making, which is used for achieving consensus and solving conflicts. And, the three layers of blockchain governance are (1) off-chain community, (2) off-chain development, and (3) on-chain protocol. Furthermore, the authors discussed which governance tasks from the six dimensions can be performed in the off-chain community, on-chain development, and on-chain layers. The framework is evaluated through expert interviews and use cases. In [78], a general overview and a mechanism for governing blockchain consortia, where multiple parties are involved in the project, were presented. This included a discussion related to setting up a blockchain consortium in a multi-party coordinated project, off-chain governance such as executing standard business practices and agreements among the partners, and on-chain governance in order to set up the blockchain and evaluate the blockchain’s progress. Ziolkowski et al. [38] discussed how the blockchain governance mechanisms of 15 different blockchain frameworks are governed. They identified six core decisions used to govern blockchains: demand management, data authenticity, system architecture development, membership, ownership disputes, and transaction reversal. Four application domains (supply chain, land registry, cryptocurrencies, intellectual property rights management) were presented as use cases to demonstrate how these core decisions can be transformed into practice. Expert interviews were conducted to study how the key decisions could be enacted in practice. In [25], three layers of blockchain governance were discussed where governance requirements can be accomplished. The three layers are off-chain community, off-chain development, and on-chain protocol. The off-chain community layer involves the execution of governance tasks in the real world, the off-chain development layer involves governance decisions regarding software development in the real world, and the on-chain layer involves governance tasks that are embedded into the blockchain protocols. In [78], two layers of blockchain governance were discussed: on-chain and off-chain. In on-chain governance, rules are hard-coded, whereas in off-chain governance, decisions are made through informal processes. On-chain governance provides the fairest, most flexible, transparent, and decentralized type of governance [78]. However, some policies are perhaps not feasible to execute on-chain such as decisions regarding protocol updates, or they are computationally intensive, so these tasks must be performed off-chain. Although off-chain governance is flexible, there are risks of error, manipulation, and transparency. An advantage of off-chain governance is that it allows for both formal and informal decisions in more flexible processes [78]. Through effective governance for blockchain, the long-term sustainability of blockchain technology can be achieved. According to the EU blockchain observatory and forum, it is important to specify who is responsible for future blockchain changes required over time and how to enforce these changes [78]. Due to the lack of an absolute governance mechanism, there have been many disputes and scandals [78,107]. As mentioned in [54,108], there is a lack of significant research in the direction of blockchain governance. Furthermore, it is not yet obvious or clearly manifested how to execute critical resolutions and enforce decrees in the blockchain [109].

## 5. Blockchain-IoT Governance

The aim of this research is to develop a decentralized, automated, and shared-value governance mechanism for blockchain-enabled IoT ecosystems, as well as utilize novel characteristics of blockchain for governance to fulfill governance requirements. Numerous governance requirements have been proposed. These requirements are discussed in the context of smart logistics as a use case, as shown in Figure 7, where every partner in the consortium can transparently verify, monitor, and configure the governance module. The proposed methodology delivers an umbrella framework for agreements in multi-party collaborations and ensures the fulfillment of the objectives of each partner. We propose a variable geometry approach for collaboration among partners and fulfillment of the governance requirements. In this section, we discuss the variable geometry governance approach, the proposed governance requirements for blockchain-enabled IoT ecosystems, and the mechanisms for fulfilling these requirements.

### 5.1. Variable Geometry Approach

There are various organizational models used to achieve collaboration among partners and coordinate policies and responsibilities at the organizational level. These include:The top-down approach, a centralized approach where a single entity manages all the other entities.The bottom-up approach, where multiple entities are involved in the decision-making process. Top-down and bottom-up approaches are unsuitable for complex scenarios [8].The variable geometry approach, a multi-level approach that is a combination of multiple mechanisms. It is a broader approach and is thus appropriate for complex [8] and heterogeneous scenarios. Complex scenarios include multiple stakeholders, varying interests, and diverse application domains where certain partners may adhere to the overall obligations while other partners may adhere to selective obligations. This mechanism is less restrictive and can potentially enhance cooperation among regional and international partners. It provides flexibility during the negotiation process [110]. We propose a variable geometry approach to implement the governance requirements.

Financial, business, or, in some cases, legal differences among partners can arise. These differences can affect the way collaborations are formed. Such differences are undesirable and can lead to conflicts. A single agreement might not be efficient. Conflicts can be resolved by accommodating the interests of partners in the consortium. Partners also have the option to opt out of conditions or policies that do not align with their financial or business interests. In the variable geometry approach, partners are not required to make the same commitments as others. This allows for less restrictive and more flexible cooperation, and such a model can enhance innovations and agile developments. The variable geometry approach offers new mechanisms to overcome negotiation challenges and potentially leads to larger cooperation [110]. It ensures flexibility in negotiation by not enforcing the complete set of obligations. The variable geometry approach [110] includes strategies to address differences in regional or multilateral agreements among partners. Partners agreeing to certain conditions should make efforts to fully comply with them. Each partner only benefits from the agreements they agree to adopt. A flexible model is required for the complex processes of negotiations. Even though the variable geometry approach is complex and difficult to design, it offers better integration of multiple entities (partners) [110]. International organizations such as the World Trade Organisation (WTO) and economic unions such as the EU have employed this approach to facilitate the negotiation process during multi-party collaboration [110,111].

The variable geometry approach offers flexibility in forming private channels among members, where certain partners could agree to certain obligations but not all. Also, channels can be formed between individual partners if they are willing to have private channels independent from the other partners. The policies and conditions are decided upon by the individual partners themselves. Channels can be formed among subsets of partners who independently agree on subsets of policies or conditions to meet specific requirements. In the variable geometry approach, stakeholders or signatories can have common policies that all the stakeholders agree on, as well as some policies for individual stakeholders who request or form specific agreements. However, parties willing to join the consortium should adhere to certain specified pre-conditions to meet the integration criteria. In private channels, certain policies might not be implemented or only implemented based on partners’ desires. This approach takes into consideration unwillingness or non-uniformity among the partners. In such a model, partners have the freedom to sign agreements based on their interests and are not necessarily required to undertake every agreement [111]. However, partners are only entitled to the benefits of the agreements they are signatories of. The variable geometry approach consists of the following common and private agreements:Common agreements are applied as pre-conditions to each partner to achieve the fundamental interests of each partner and the collective aim of the consortium. For instance, all the partners agree on privacy and security requirements.Private agreements are agreed upon among partners while forming channels to provide certain services. Multiple stand-alone agreements can be signed, where every contracting party is free to join any agreement they want. For instance, certain partners can agree among themselves on a specific cost model for a service.

Blockchain technology is comprised of distinctive frameworks, and thus it is exceptionally suited to the variable geometry approach. Enterprise blockchains have a different paradigm compared to public blockchains [112] because enterprises are profoundly discreet about business data and processes [112]. For instance, in a public blockchain, it is expected that every transaction is public; however, in an enterprise blockchain, businesses are unwilling to make all transactions public for various reasons such as business secrets and competition [112]. Therefore, certain blockchain frameworks allow for the forming of private channels among partners, such as Hyperledger Fabric [73].

### 5.2. Proposed Governance Framework

We demonstrate specific proposals to establish a legal framework for blockchain-enabled IoT ecosystems, which include the requirements detailed below. It is essential to address these requirements in a blockchain-enabled IoT governance framework. These requirements lay the foundation for a comprehensive governance mechanism and provide significant value to the consortium (collaboration) and services. Where possible, we further categorize these requirements into sub-categories, as shown in Figure 9. Furthermore, the proposed mechanism is highly flexible; the proposed requirements can be adopted according to the individual needs of the consortium. Table 1 summarizes the requirements for implementation in off-chain or on-chain development.

#### 5.2.1. Purpose

At the time of consortium (collaboration) formation, the purposes for the adoption of a blockchain-IoT system need to be determined. Purposes can be subdivided into three categories; problems, vision, and roles. For instance, what problems can a blockchain-IoT system potentially solve, will a blockchain-IoT system be able to effectively solve these problems, and what value will the system return or generate. After initial agreements about the value a blockchain-IoT system can add to the consortium, a formal organization can be formed with all the partners, who agree on the initial goals of the project. A dedicated team is required to decide upon the intellectual property rights. The roles, which include actors and their responsibilities, are determined, such as who supports which values and who contributes what. The private and public sectors have different roles and responsibilities. The role of each actor or partner needs to be clearly identified and agreed upon. It is important that each partner interrelate and coordinate with all the other partners [78]. Roles might include international, national, and regional actors. Clear policies regarding participation in the blockchain network, for instance, who can execute transactions and which nodes are allowed to read the ledger (data) and identify validators (miners) [78]. The agreements are clearly compiled in order to avoid future conflicts. The over-regulation of the technical environment should be avoided, as it can cause unnecessary burdens [8] and potentially lead to limitations on technical innovation. This requirement can be determined during off-chain development and implemented during on-chain development. On-chain policies are required regarding user authorization, access to data, and achieving and maintaining transparent consent among users. This requirement is typically included in the common agreement of the variable geometry approach.

#### 5.2.2. Ethics

As per the EU Commission, there are mainly six ethical issues: social justice and (digital) divides, trust, blurring of contexts (private vs. public), non-neutrality, agency (social contract between people and objects), autonomy (informed consent vs. obfuscation of functionality) [113], and corporate social responsibility in enterprises [8]. Therefore, it is important to raise ethical awareness among people who are part of the blockchain-IoT consortium to ensure consent and fairness and avoid the introduction of backdoors and the exploitation of users. This is especially important for developers, auditors, regulators, and stakeholders. Some of the ethical considerations that should be included in ethical policies are personal identity, autonomy of individuals, user consent, fairness, and social justice [8]. The development of adequate policies to enforce ethical aspects in the design and development of IoT solutions [8] and blockchain technology is vital. Awareness among citizens leads to the integration of ethics in technologies to some extent. All partners should agree to uphold ethics. The distinctive features of blockchain can be utilized to adhere to ethical principles. The ethical mechanisms mentioned above should be decided and agreed upon during off-chain development and implemented and verified during on-chain development. This requirement could be included in the common agreement of the variable geometry approach.

#### 5.2.3. Transparency

Transparency is an integral part of ethics and should be embraced when developing governance policies [8]. Principles regarding transparency are crucial for gaining solid adoption since blockchain-IoT technology is still in its infancy. The framework should elaborate on which data to use, how to obtain user consent, how to process and store data, why to collect and use data, who controls the data, data deletion, and how to ensure data transparency to establish trust with users. The framework should include policies that are legitimate and fair, upholding the democratic principles of society. Blockchain offers all these features of transparency. Various existing technological impediments (i.e., transparent tracking, validation, and recording) can be addressed by using blockchain. Transparency can be achieved through the immutability, traceability, and consensus characteristics of blockchain. The choice of consensus mechanism can affect the security and balance of the blockchain [78]. There should be consequences for any wrongdoing by any entity. The consequences should be clear for the accountability bodies to impose and apply them fairly. Transparency requirements can be developed during on-chain development and could be included in the common agreement.

#### 5.2.4. Audit

An audit is a process for continuously monitoring the purpose and scope of controls, such as analyzing whether controls are functioning as required and whether they are applied to systems or processes that are part of the scope. The purpose of an audit is to eradicate any fault, risk, or vulnerability in the controls. Some auditing features are as follows:*Accuracy*: Assesses the controls.*Completeness*: Are the controls adequate, are there gaps, and are the controls applied thoroughly.*Timeliness*: The controls are executed on time.*Resilience*: The controls are resilient to failures and there are backups if the primary controls fail.*Consistency*: The primary and secondary controls are correctly in place.

An audit can be performed both on-chain and off-chain. Off-chain auditing is conducted to monitor an individual partner’s contributions and commitments. In on-chain auditing, technical tools and mechanisms are put in place to monitor performance. Automated audit mechanisms based on blockchain could be implemented in order to audit common agreements, as well as private agreements (channels).

#### 5.2.5. Interoperability

As per the Internet Engineering Task Force (IETF) definition, the Internet is “a large, heterogeneous collection of interconnected systems that can be used for communication of many different types between any interested parties connected to it” [52]. The Internet consists of the “core Internet”, that is, Internet service provider (ISP) networks, and the “edge Internet”, which comprises private and corporate networks [52], proprietary and off-the-shelf software, and frameworks. For collaboration and cooperation, interoperability (platform-independent solutions) across various partners, architectures, and domains is vital. For instance, an existing single market such as the European Union has invested significantly in fostering interoperability at various levels [114] in order to have a uniform digital identification in the union. Policies have been adopted in the European Interoperability Framework (EIF) and European Interoperability Reference Architecture (EIRA) to incorporate interoperability [114]. The framework should include mechanisms for interoperability both at the network and architecture levels and, more importantly, because of the data exchange that occurs nationally and internationally among the partners. Interoperability can be applied to various aspects such as legal, organizational, semantic, and technical [114]. Efficient and effective interoperability policies can facilitate interactions among partners [114]. Key stakeholders are involved in decisions regarding which network architectures and technological strategies to use in order to maintain interoperability. Ensuring interoperability among IoT ecosystems and with systems that are outside the blockchain network perpetuates the scalability of diverse networks. Interoperability guarantees that existing systems are adaptable to new innovations [8]. One of the main IoT challenges is the heterogeneity of IoT devices. Different IoT devices have varying compatibilities, support different protocols, and have different computational capabilities, which makes interoperability very complex. One way to overcome this is to embrace or incorporate existing standards and governances (IETF, ICANN, Internet, RIRs, ISOC, IEEE, IGF, W3C, cloud governance, IoT reference architectures like Industrial Internet Reference Architecture (IIRA), Internet of Things Architecture (IoT-A), and IEEE P2413) into the framework in order to leverage their advantages. At the organizational level, the framework should incorporate mechanisms for forming good relations with other national and international bodies. Interoperability requirements can be implemented during off-chain (organizational and legal level) and on-chain development. This requirement could be included in the common and private agreements of the variable geometry approach.

#### 5.2.6. Architecture

There are three main types of architecture: centralized, decentralized, and distributed. There are pros and cons associated with each of these architectures. For instance, in a centralized architecture, it is hard to achieve privacy, and networks based on a centralized architecture can suffer a single point of failure [8]. In the IoT infrastructure, availability is crucial for service providers [8]. Availability can be improved or made more robust with a distributed architecture [8]. IoT architectures should be able to cope with large numbers of users, and there should be no restrictions on the number of users. Also, mechanisms should be put in place to reduce delays in the communication of information between users and systems or devices [8]. Blockchain technology is a better alternative to existing centralized technologies. So, incorporating blockchain technology into the IoT ecosystem could be highly beneficial. Furthermore, any blockchain framework that is incompatible with the IoT architecture can potentially hinder blockchain adoption in the IoT ecosystem. Architecture requirements can be implemented during off-chain and on-chain development. This requirement could be included in the common and private agreements (channels) where partners can form channels for their own requirements.

#### 5.2.7. Security

Security is one of the crucial governance requirements for various reasons. Mechanisms should be developed to ensure confidentiality, integrity, and availability. The framework should incorporate strategies to ensure security at every level of the infrastructure (devices, networks, integration, and physical). Mechanisms should be put in place for continuous security hardening. Security requirements can be further categorized into sub-requirements such as physical security, end-to-end security, security best practices, security risk assessment, real-time intrusion detection and prevention, enhanced identification and authentication, and security audits. Various security mechanisms have been proposed in the literature. Almeida et al. [1] recommended four principles be incorporated when deploying IoT applications in order to secure users’ data and build trust in the IoT. These are notice and choice, data minimization, access to personal data, and accountability. IoT devices are required to adhere to reasonable security requirements; for instance, the inclusion of mechanisms for encryption, authentication, and access control to ensure that user identification cannot be traced back. Various anonymity mechanisms such as ZCash anonymity and monero anonymity [78] have been proposed. Most of these security requirements can be achieved through blockchain technology. Security requirements can be implemented mainly during on-chain development. However, in cases where physical security is required, it can be implemented during off-chain development. This requirement could be included in the common and private agreements of the variable geometry approach.

#### 5.2.8. Privacy

Various definitions and types of Personal Identifiable Information (PII) and Sensitive Personal Information (SPI) are present in the literature [115,116]. Centralized servers owned by third parties have the capability to access, monitor, and manipulate users’ data since digital technologies can be used to discriminate or track users’ behavior. Therefore, ensuring privacy can increase confidence in the technology and ultimately the business’s growth. Since ensuring privacy is one of the key challenges in governance frameworks, service providers should take care of human integrity, identity, and privacy when providing services [8]. Individuals should have full authority and control over their data (personal, financial, commercial, etc.). The framework should formulate policies regarding the security of data against unauthorized access, limit data collection and data dissemination, and determine who has authorized access to the data. Perhaps the governance framework should ensure compliance with existing European data protection laws such as the data protection directive (95/46/EC) [117], ePrivacy directive (2002/58/EC)(2009/136/EC) [118], and General Data Protection Regulation (GDPR) [34]. The GDPR [34] presented six requirements regarding user data processing, including lawfulness, fairness and transparency, purpose limitations, data minimization, accuracy, storage limitations, integrity, and confidentiality [119]. These laws ensure the protection of users’ rights, protection of users’ data, transparency, and accountability by proposing various privacy paradigms such as privacy by default, privacy by design, and privacy as confidentiality. Where necessary, the data should be anonymized to avoid personal identification; for instance, the patterns of energy consumption of certain households can potentially be used to track their behavior or lifestyle. This is because privacy is not a static entity; it varies with the application domain and the way data are collected, processed, and stored. Other strategies and recommendations for building strong data privacy schemes are mentioned in [120]. Cryptographic mechanisms, such as homomorphic encryption, k-anonymity, data obfuscation, Zero-Knowledge Proof (ZKP), secure multi-party computation (SMPC), and ring signatures, can be used to implement privacy and anonymity [121]. Most of these privacy requirements can be achieved through privacy-preserving solutions for blockchain technology mentioned in the literature, e.g., [119]. Distributed ledger technology has the technological capabilities to minimize privacy exploitation to some extent. Users can view the type of data that are collected and stored. Blockchain technology can solve privacy issues using various cryptographic mechanisms and through immutable, transparent, and distributed ledger characteristics of the technology. However, different blockchain frameworks impart different levels of privacy [121]. Various mechanisms were demonstrated in [121] to achieve privacy in blockchains. Privacy requirements can be implemented during on-chain development and they can be included in the common and private agreements (channels).

#### 5.2.9. Fault Tolerance

Fault tolerance is important for the availability of blockchain-IoT services and it is closely related to security since it improves security, trust, and performance. Fault tolerance is one of the major challenges in distributed systems since networks and systems are prone to faults, errors, and failures. Faults might arise due to various reasons such as hardware, software, network, or system failures due to malicious errors. The framework should determine the fault tolerance mechanisms for the blockchain-IoT system so that the networks and systems are reliable in cases of fault or errors and avoid random downtime. Although fault tolerance mechanisms depend on the architecture of the network, five common fault tolerance phases are suggested, which are fault detection, fault diagnosis, evidence generation, assessment, and recovery [122]. Various static and dynamic data replication schemes are discussed in the literature in order to improve system availability [123]. Although public blockchains are based on a peer-to-peer network architecture, they have inbuilt capabilities to cope with faults. For instance, even if only a few nodes in the network are working correctly, the blockchain can function. Therefore, it is beneficial to integrate blockchain and IoT technologies. However, fault tolerance mechanisms are required, especially for software level and integration failures. Enterprise blockchain frameworks allow limited and authenticated nodes to participate in the consortium, and channels are formed among the enterprises. So, the failure of a node in an enterprise blockchain can have an adverse impact on the consensus since each enterprise is represented by its node, and the failure of a node can, consequently, lead to the failure of the blockchain [112]. Any downtime in the enterprise blockchain can be very costly [112]. Therefore, efficient fault tolerance mechanisms are essential in order to ensure the smooth functioning of an enterprise blockchain and make the infrastructure resilient to failures [112]. Also, different blockchain frameworks have varying fault-tolerance capabilities. Podgorelec et al. [124] discussed and compared the fault-tolerance capabilities of Hyperledger Fabric and Iroha, which are based on the performance of consensus mechanisms. Fault-tolerance requirements can be implemented during on-chain development included in the common and private agreements.

#### 5.2.10. Performance Measurement

As mentioned earlier, collaborations are formed among the partners in order to provide various services to customers. It is important to regularly evaluate and determine the framework or business performance to ensure that it is beneficial to both users and stakeholders. Business requirements are evaluated against the current status, expectations, and future policies and, consequently, strategies are determined. Performance metrics are established in order to keep track of success in the form of customer satisfaction and return on investment (ROI) and plan future improvements accordingly. Performance measurements include various key performance indicators (KPI) to evaluate the operations of service delivery [125]. Performance metrics can be used to contemplate the framework’s capabilities, effectiveness, and fruition [125]. Various criteria such as return on investment (ROI) and total cost of ownership (TCO) are used to evaluate projects’ performance; however, these two are not the only metrics [125]. For instance, Grembergen and Haes [126] presented a balanced scorecard approach to measure performance. In this approach, various perspectives and objectives were demonstrated. Various types of performance metrics were discussed in [125], including process metrics, service metrics, enterprise goals, and sample metrics. Therefore, stakeholders can use a more comprehensive cost-benefit analysis performance measurement approach that is based on quantitative and qualitative indicators [125]. Performance measurement requirements can be implemented during off-chain development and included in the common and private agreements (channels).

#### 5.2.11. Cost

To maximize profit, partners require a suitable business model, along with fair pricing strategies, to avoid conflicts and ensure that the price is suitable for the blockchain-IoT ecosystem. Pricing mechanisms include rules regarding charging users or stakeholders for services. The cost model or pricing strategy plays an important role in collaboration, as it is used to identify the business interests of partners, and a fair pricing model can potentially attract a large number of users. Various pricing mechanisms and policies have been discussed in the literature [127,128]. Each pricing strategy has advantages and disadvantages. However, certain cost models are a good match for an IoT-blockchain system; for instance, models that are affordable, flexible, and predictable. Deciding on cost models is always challenging since there are various partners involved and there are various types of cost models. Some of the challenges of pricing are unpredictability, fairness, and making pricing enticing to users. Because different partners or users might have different usage patterns, different services might cost different amounts, and some users might choose only selected services. Some of the well-known pricing models are the pay-per-use and pay-per-device models. In the pay-per-use model, users pay for the service according to the amount of usage of the service. Such a payment model is efficient and straightforward since the user pays only for the duration or amount they require. This model can be based on the duration of usage, data flow, or power usage. In the pay-per-device model, users pay per device service. For instance, a fixed amount per month per IoT device. Other possible strategies are storage as a service (SaaS), software as a service (SaaS), monthly or yearly support costs, and up-front charges. Cost requirements can be determined during off-chain development, implemented in on-chain development, and included in the common and private agreements (channels).

#### 5.2.12. Scalability

As mentioned earlier, blockchain for the IoT enhances the trust and security of the IoT. However, as the number of IoT devices is increasing at a very fast pace, the lack of blockchain scalability and compatibility capabilities can be an obstacle. The transaction rate of blockchain-based technologies is significantly slower compared to existing digital transaction systems [129,130]. For instance, the number of Ethereum transactions per second (TPS) is 15–20, and the average transaction confirmation time is 2 min. The number of Bitcoin transactions per second is 3–7 and the average transaction confirmation time is 25 min [129]. The lack of efficient scalability mechanisms can impact real-time transaction validations and adversely impact the large-scale adoption of blockchain. Therefore, the integration of blockchain and the IoT requires efficient mechanisms to enhance scalability. Regulations are required regarding the size and duration of data storage, transaction validation time, compatibility, and interoperability. Various mechanisms have been proposed in the literature to improve blockchain scalability. For instance, Hazari and Mahmoud [129] proposed a parallel proof-of-work mechanism to improve blockchain scalability. Boyen et al. [131] proposed a similar parallel mining mechanism in order to execute transactions swiftly. Other mechanisms for improving scalability include enhancing network latency, reducing transaction queuing, enhancing compatibility between private and public blockchains, enhancing compatibility between the IoT and blockchain, and simplifying the complexity of smart contracts. Scalability requirements can be implemented during on-chain development and included in the common and private agreements (channels).

#### 5.2.13. Automation

Blockchain and automation are highly related. With the increase in the number of IoT devices and consequently, the increase in complexity, automation mechanisms are required to improve efficiency and speed without compromising transparency and traceability. Blockchain is deployed in many application domains, such as supply chains, trust building, and workflow management, to achieve automation. Blockchain features a distributed architecture and automated execution of transactions without the interference of third parties when certain conditions are met (smart contracts), which is highly beneficial, as these features avoid the threats associated with centralized architectures and third-party interference. Efficient mechanisms for the automated execution of transactions are required to improve productivity and performance, save costs, and avoid errors. Smart contracts can play an important role in the automation of various tasks in many diverse application domains. Scalability requirements can be implmented during on-chain development andincluded in the common and private agreements (channels).

#### 5.2.14. Sustainability

The Internet is typically managed by various organizations such as the Internet Engineering Task Force (IETF), Internet Research Task Force (IRTF), Internet Engineering Steering Group ((IESG), Internet Corporation for Assigned Names and Numbers (ICANN), Internet Architecture Board (IAB), and World Wide Web Consortium (W3C). Each of these organizations has varying responsibilities and purposes. The modern world is heavily dependent on Internet sustainability and it is a critical resource for the mankind [52]. Therefore, international law is required for the protection of this critical resource [52]. Blockchain technology has the potential to provide long-term sustainable services and infrastructures such as automated compliance checks and integration with the IoT [132], meeting the goals of the 2015 Paris climate agreement [133] and eliminating corruption as per the United Nations Environment Programme (UNEP) 2030 goals [134]. However, blockchain is mainly dependent on heavy computational operations that require significant power, which is highly concerning for sustainability. Blockchain applications that consume a high amount of energy are not environmentally sustainable [133]. The governance should consider the long-term environmental and economic sustainability of the blockchain technology by reducing energy consumption while not compromising security. The blockchain framework should be capable of effectively evolving in order to adapt, change, and interact with environmental requirements. Sustainable blockchain mechanisms have been mentioned in the literature [135]. Sustainability requirements can be implemented during on-chain development and included in the common and private agreements (channels).

#### 5.2.15. Support

Since users are an integral part of a business, support requirements play an important role in the success of the business. It is possible to retain and attract more customers through user-friendly customer support mechanisms. The framework should adopt different strategies and methodologies regarding measuring and valuing customer satisfaction and meeting users’ expectations. There are various types of support including organizational support, technical support, and legal support, which includes legal obligations and legal compliance. Support can be remote, physical, or in the form of documentation. Regular evaluations should be carried out regarding customer satisfaction with services, and policies and strategies should be developed accordingly. Various customer satisfaction and monitoring strategies were demonstrated in [136]. Support requirements can be implemented during off-chain and on-chain development and included in the common and private agreements (channels).

## 6. Proposed Governance Framework Use Cases and Evaluation

In this section, we demonstrate a smart logistics use case and apply our governance mechanism to the use case in order to analyze the feasibility of the proposed framework. In the smart logistics use case, we elaborate on the interests of each partner in the context of smart logistics. Then, we discuss how to achieve these interests through our proposed governance framework. Furthermore, as shown in Figure 10, we set up a two-node Ethereum [83] blockchainconsisting of a standard voting mechanism. Ethereum [83] is one of the largest public blockchains with respect to market capitalization [25], and it supports smart contracts, which we can use to implement the governance requirements. Solidity is used to develop the voting mechanism. The source code is available at [137]. The sole purpose of this setup is to demonstrate the feasibility of integrating the IoT (Raspberry Pi) and blockchain.

### 6.1. Smart Logistics Use Case

There are many use cases of enterprise blockchains such as financial agreements and transactions, smart contracts, records management and data sharing, smart grids, identity management and authentication, e-voting, and smart logistics [112]. Smart logistics include a large number of distributed sensors, radio, and other technologies. Let us consider a smart logistics scenario, in which a product such as coffee is being transported from South America to Europe. There are multiple partners involved including suppliers, manufacturers, distributors, government, retailers, consumers, and other partners, as shown in Figure 8. The supplier supplies raw materials, the manufacturer produces the product, and the distributor distributes the product to the consumers. Suppliers, manufacturers, distributors, and retailers are paid for their efforts. Governmental bodies are involved in implementing the national protocols and laws for smart logistics. Other national and international partners might also be involved in the distribution. Collaboration among these partners is crucial for high-quality smart logistics services. Each partner in the consortium has their own needs and obligations. The supplier is expected to provide high-quality coffee. Consumers want to receive the product in good condition, avoiding any damage or loss of goods and ensuring transparent information about the product’s origin. Distribution can be carried out through road, sea, or air transport, and the distributor ensures minimal delays and provides transparent proof of delivery. The partners expect ethical policies regarding social justice, identity, autonomy, consent, and fairness. The consortium ensures compliance with agreements, commitments, and liabilities in case of non-compliance. To achieve collaboration among diverse partners, the consortium supports interoperability since it may include heterogeneous devices, networks, and frameworks. Since smart logistics might consist of a large number of devices, scalability and a distributed architecture can be crucial. All partners are concerned about confidentiality, integrity, and availability. The overall purpose of the consortium is to provide excellent smart logistics services and financial gains for the partners.

### 6.2. Proposed Governance Framework Evaluation

In this section, we demonstrate the feasibility of the proposed framework by applying it to the smart logistics use case mentioned earlier. In order to fulfill the requirements of the smart logistics use case, we propose a blockchain-based architecture for smart logistics, as shown in Figure 7, and apply the proposed governance framework. For simplicity and clarity, we follow a two-layer (on-chain and off-chain) governance to decide upon and implement the proposed framework requirements. Blockchain technology has the technical capabilities to provide innovative, transparent, and distributed smart logistics services. According to the proposed framework, the partners initially identify the smart logistics problems that can be solved using blockchain. Furthermore, policies are defined regarding the vision of the consortium, where the objectives of each partner are guaranteed. Prior to developing smart logistics services, the concerned stakeholders are required to be consulted and convey their concerns. According to the vision, roles and responsibilities are assigned to each partner. In the smart logistics use case, the vision can be to provide innovative services and increase profit. The roles might include smart logistics partners, technical service providers, developers, and national and international governmental bodies. Technical service providers can develop and maintain blockchain. Governmental bodies are responsible for ensuring compliance with national and international regulations. Innovative principles are included in order to promote investment and financial gains and assign roles accordingly. Policies and frameworks are in place to ensure users’ consent and that the consortium functions as agreed. As per the proposed framework, developers, auditors, regulators, and stakeholders are instructed to ensure the solutions and services are developed according to ethical principles. In case of a violation of users’ ethical rights, the consequences should be straightforward for the accountability bodies to fairly impose. During data collection and processing, transparent consent should be implemented, and users should have full control over their data. To avoid unnecessary delays, fast and efficient transaction mechanisms should be developed. Automated audit mechanisms, where each partner can verify the ledger, should be provided, since in blockchain technology, each partner has a copy of the ledger, so it is possible to independently verify it. Comprehensive policies for facilitating and maintaining collaboration and cooperation among the various partners should be developed, and interoperability among the partners at the technical, legal, and organizational levels should be ensured, which can attract more potential partners and consumers. The consortium should implement novel security and privacy mechanisms, and in order to fully leverage the blockchain features, a distributed architecture should be implemented. A distributed architecture has many applications and includes security, transparency, trust, privacy, availability, and inbuilt capabilities for coping with errors and faults. Furthermore, existing privacy-preserving mechanisms and standards should be adopted. The smart logistics consortium should develop fair and economical cost strategies that are beneficial to each partner. Furthermore, services, user satisfaction, and economic gains should be regularly evaluated, and services and strategies should be updated accordingly to attract and retain customers. So, in conclusion, following these requirements would transform traditional logistics into efficient and profitable smart logistics (Table 2, Figure 9 and Figure 10).

## 7. Conclusions

In this research, we have presented a governance framework for blockchain-enabled IoT applications, comprising various essential requirements. Furthermore, we have demonstrated that most of these requirements can be met through blockchain technology due to its unique characteristics, thereby enhancing the security and reliability of the IoT. The IoT generates a tremendous amount of data, and its growth has resulted in increased architectural complexity, leading to various security threats. Data processing methods are becoming increasingly automated and intelligent, enabling the precise learning and prediction of distinct human behaviors. With recent advancements in various industries, products have become smarter, more efficient, and safer. However, they have also become more intrusive. According to European data protection regulations, IoT devices can be directly linked to individuals through wearable devices, quantified self-tracking devices, and robotics with sensors used in home automation. One of the main challenges is the lack of a dedicated governance mechanism. Without a governance framework, any technology can potentially become dubious and invasive. Compliance frameworks are implemented to safeguard sensitive data, ensure accountability, and more. Governance encompasses policies, roles, and the enforcement of the rule of law. Through governance and legal frameworks, it becomes possible to regulate both the commercial and technical aspects of the technology, facilitate further development, attract investors, and improve collaboration, ultimately gaining consumers’ trust in the technology. Existing governance frameworks often lack the fundamental requirements of IoT and blockchain technologies. There are numerous underlying governance challenges that need to be addressed. For instance, in IoT governance, the challenges include legitimacy, transparency, accountability, anti-competitive behavior, varying organization sizes, heterogeneity, ethics, privacy, security, competition, and the resolution of complex disputes among partners. As mentioned earlier, blockchain can be used to mitigate most of these IoT challenges. For example, unlike traditional logistics, smart logistics has brought much-needed efficiency. However, smart logistics is far from perfect and still encounters many challenges. One potential approach to overcoming these challenges is to adopt blockchain technology. Therefore, efforts are required to promote blockchain technology within the IoT ecosystem. Blockchain allows for transparent obligations within agreements and clear consensus mechanisms to verify adherence to governance standards. The proposed framework comprises 15 requirements for a blockchain-enabled IoT ecosystem. Each of these requirements was briefly studied. Arguments were presented to underscore the importance of each requirement, and mechanisms were detailed for achieving them. These requirements were designed to address the numerous challenges and work involved in making the blockchain-IoT system a mainstream technology. We have researched these requirements while taking into account the economic, social, and environmental incentives and interests of every partner within the blockchain-IoT consortium. Our proposed governance mechanism also considers the need for a distributed IoT ecosystem. Furthermore, we advocate for a variable geometry approach in implementing the requirements to promote flexibility, allowing partners the freedom to opt out of agreements that are not relevant to their interests. Finally, we have evaluated the framework through a smart logistics use case.

## 8. Future Work

We have discussed the requirements, challenges, and potential enhancements from a broad perspective. Future research should delve deeper into each of these requirements and propose concrete mechanisms to address them. Additionally, conducting a survey involving multiple companies to evaluate the framework and gather recommendations for refining existing requirements or introducing new ones would be beneficial.

## Figures and Tables

**Figure 1 sensors-23-09031-f001:**
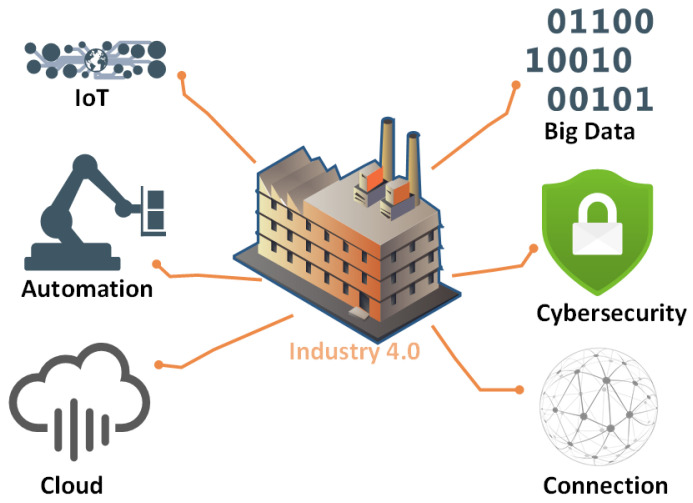
Industry 4.0 revolution.

**Figure 2 sensors-23-09031-f002:**
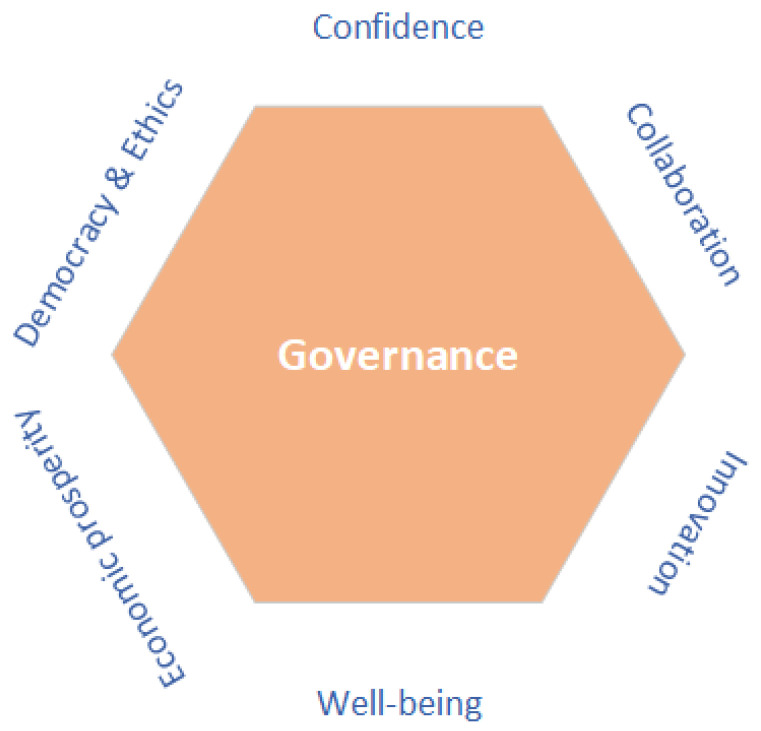
An illustration of the main cogitation principles of governance.

**Figure 3 sensors-23-09031-f003:**
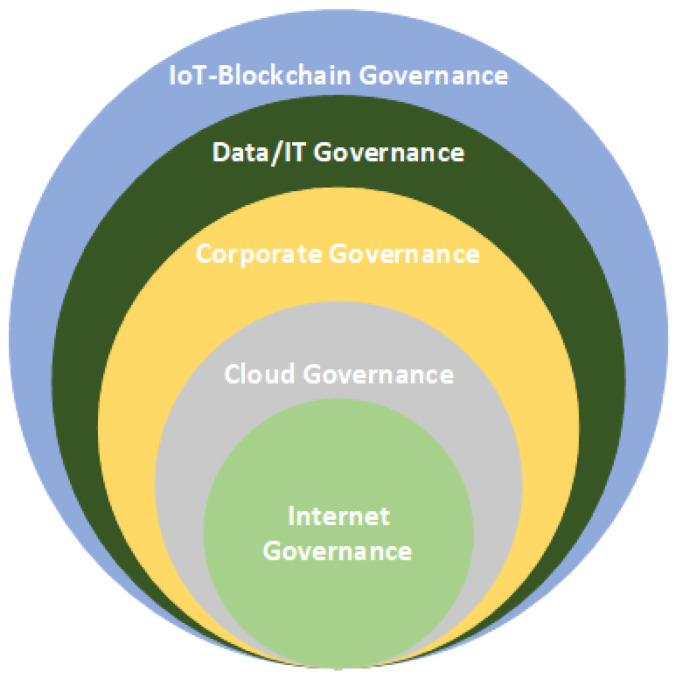
Some of the well-known governance frameworks. A blockchain-IoT system requires more comprehensive policies and roles; therefore, it includes policies and roles from existing governance frameworks.

**Figure 4 sensors-23-09031-f004:**
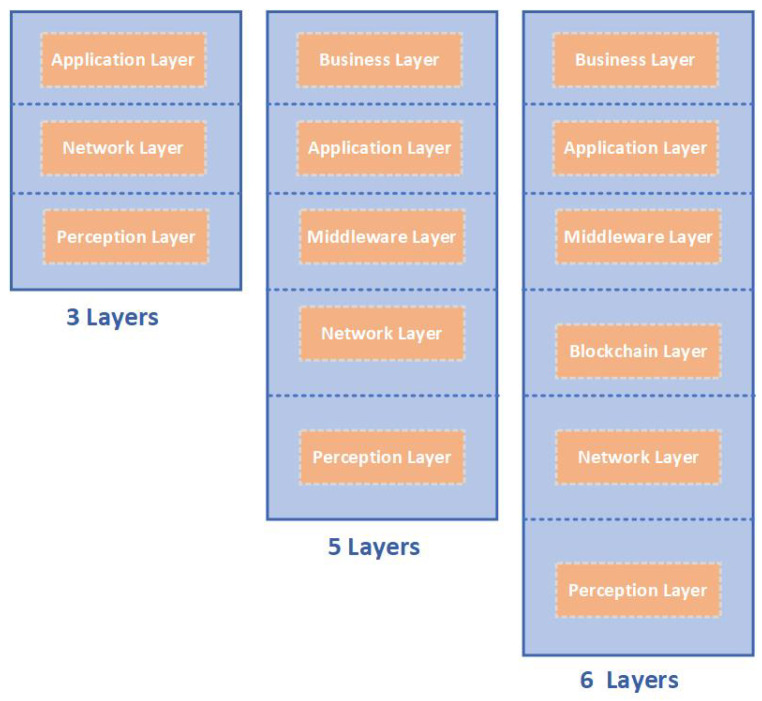
Six-layer IoT reference architectures for a blockchain-enabled IoT ecosystem.

**Figure 5 sensors-23-09031-f005:**
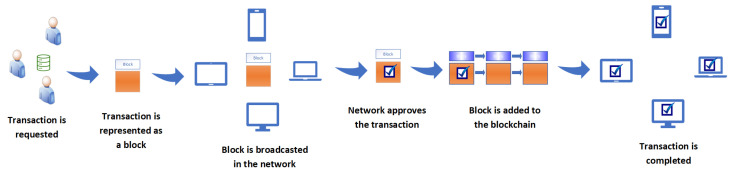
Mechanisms of blockchain.

**Figure 6 sensors-23-09031-f006:**
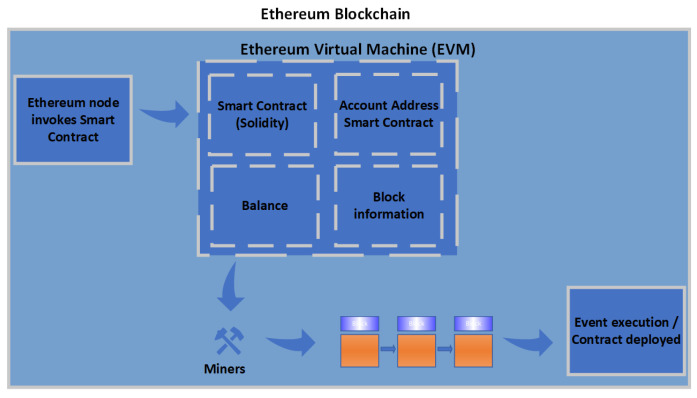
Mechanisms of Ethereum Virtual Machine (EVM).

**Figure 7 sensors-23-09031-f007:**
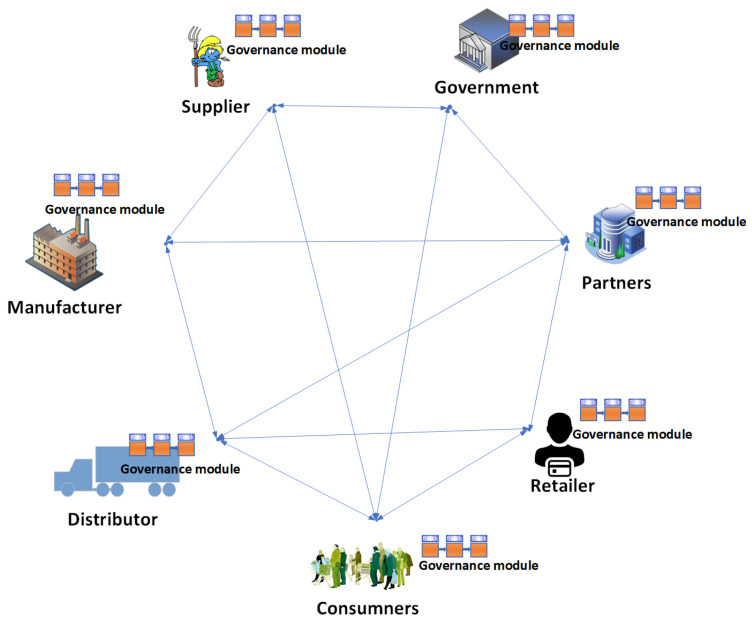
Distributed governance framework for blockchain-enabled smart logistics.

**Figure 8 sensors-23-09031-f008:**
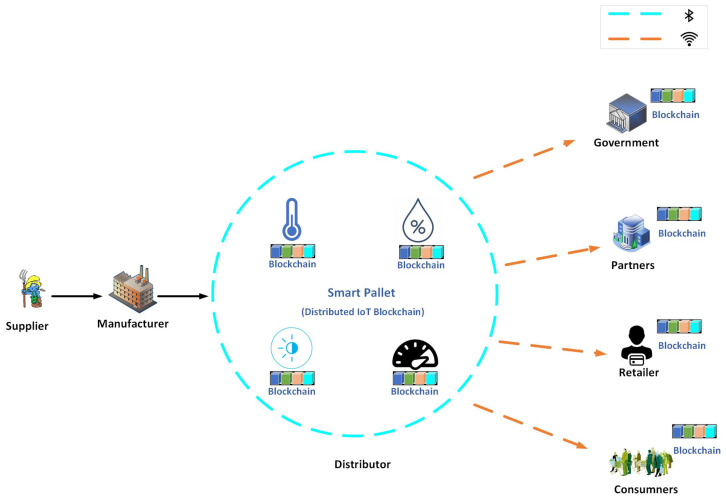
An illustration of an ideal blockchain-enabled smart logistics scenario. We evaluate our proposed mechanism in a smart logistics use case.

**Figure 9 sensors-23-09031-f009:**
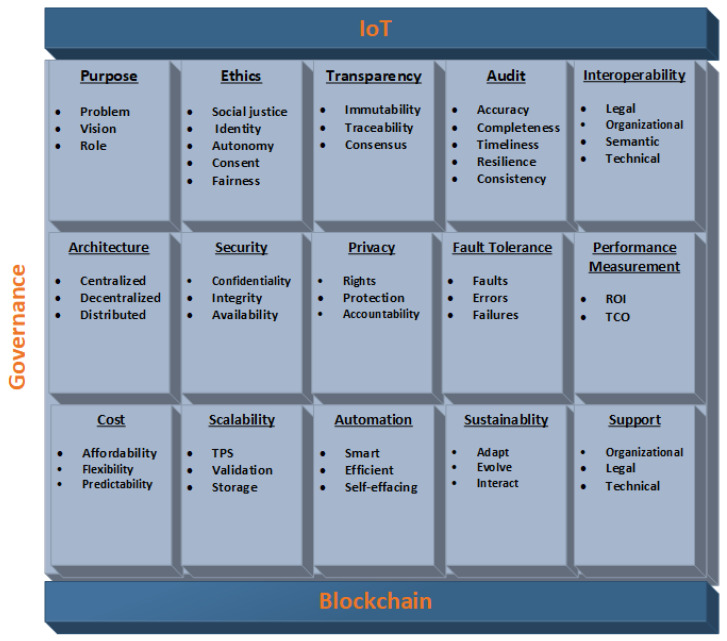
Framework. Governance specifies how roles are assigned and how updates are executed.

**Figure 10 sensors-23-09031-f010:**
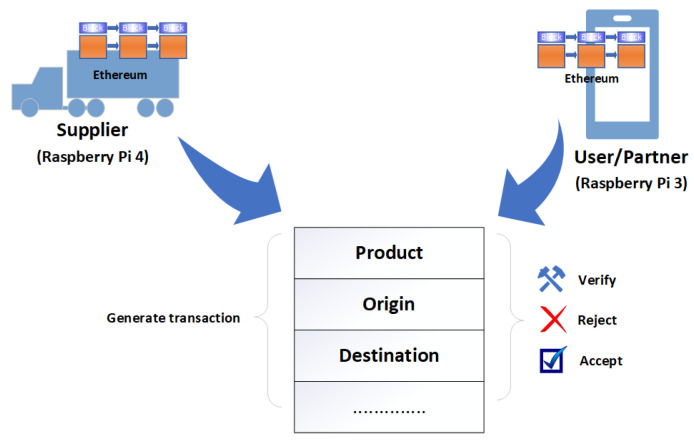
Transaction execution in blockchain-enabled smart logistics.

**Table 1 sensors-23-09031-t001:** The table shows the governance requirements that can be implmented off-chain and on-chain. For simplicity and clarity, we follow a two-layer (on-chain and off-chain) governance.

Requirement	Off-Chain	On-Chain
Purpose	✓	✓
Ethics	✓	✓
Transparency	-	✓
Audit	✓	✓
Interoperability	✓	✓
Architecture	✓	✓
Security	✓	✓
Privacy	-	✓
Fault tolerance	-	✓
Performance measurement	✓	-
Cost	✓	✓
Scalability	-	✓
Automation	-	✓
Sustainability	-	✓
Support	✓	✓

**Table 2 sensors-23-09031-t002:** Specifications of the IoT devices used in the experiment.

Component	Specifications (Raspberry Pi 4)	Specifications (Raspberry Pi 4)
Model	Model B	Model B
Processor	Broadcom 2711, Cortex-A-72, 64-bit SoC @ 1.5 GHz	Broadcom 2711, quad-core Cortex-A72, 64-bit SoC @ 1.5 GHz
Internal Working Memory (RAM)	4 GB	8 GB
SD card support	Micro SD card (operating system and data storage)	Micro SD card (operating system and data storage)
SD card size (capacity)	64 GB	128 GB
Connectivity	(a) 2.4 GHz and 5 GHz IEEE 802.11.b/g/n/ac wireless LAN (b) Bluetooth 5.0, BLE (c) Gigabit Ethernet	(a) 2.4 GHz and 5.0 GHz IEEE 802.11b/g/n/ac wireless LAN (b) Bluetooth 5.0, BLE (c) Gigabit Ethernet
Operating system	Raspbian	Raspbian

## Data Availability

Data are contained within the article.

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
