# Peer review of "Governance of a Blockchain-Enabled IoT Ecosystem: A Variable Geometry Approach"

_sensors, 2023, doi:10.3390/s23229031_

Round 1

Reviewer 1 Report

Comments and Suggestions for Authors

. In this paper, the authors address the role of blockchain in governance an propose a novel governance framework for blockchain-enabled IoT ecosystem.

. Overall, the paper is well structured and organized, with good logical flow.

. In the Introduction and related work should be expanded to include important recent references on the application of blockchain and smart contracts in different domains including specifically in healthcare IoT:

https://www.sciencedirect.com/science/article/pii/S0360835222000651

. The paper should address governance for the different ttypes of BC and DLT – specifically Hperledger and Ethereum Besu private bockchains.

Comments on the Quality of English Language

Can be improved. But readable. 

Reviewer 2 Report

Comments and Suggestions for Authors

Dear Authors:

I revised the manuscript: “Governance of Blockchain-enabled IoT Ecosystem: The Variable Geometry Approach” submitted to the “Sensors-MDPI” Journal. Though I have doubts about the paper's novelty the paper is very interesting. However, my conclusion is that this paper requires a significant amount of improvement to get published.

Article topic:

The theme of the article is not concise and does not accurately reflect the content. Consider revising both the theme and the structural organization to enhance the overall quality of the paper.

Overall Comments:

1.       The manuscript does not adequately highlight the author's contributions. The novel aspects and contributions of the research are not clearly presented in the paper, especially in the Introduction section.

2.       Evaluation of the proposed governance framework is inadequate. Some more technical and relevant scientific evaluation techniques should be represented.

3.       Some of the subsections seemed redundant. For example, subsection: 3.2. Role of blockchain in governance and subsection: 4.2. Role of blockchain as governance mechanism can be considered as same except some references.

4.       The organization of the manuscript is satisfactory, but the result discussion of the proposed governance framework needs more improvement. Description of Table:2 is missing.

5.       The manuscript lacks a comparison with relevant works in the field, which is essential for providing context and demonstrating the novelty of the research. Please incorporate a comparative analysis with relevant literature to strengthen your paper.

6.       How can agent-based framework assist in Governance of Blockchain-enabled IoT Ecosystem.  You can review some references such as Device Agent Assisted Blockchain Leveraged Framework for Internet of Things (2022).

Comments on the Quality of English Language

Authors should work on the language so that the contents are more readable. 

Round 2

Reviewer 2 Report

Comments and Suggestions for Authors

1. The reviewer is suggesting providing an outline figure of the contents of this article. The author can follow Figure 18 in the following article: https://www.sciencedirect.com/science/article/pii/S2096720921000014

2. The current challenges of adopting blockchain in IoT ecosystem governance and future research directions in this field should be presented in a Table for increasing the readability of the important information in the paper.  

Comments on the Quality of English Language

Try to improve the readability of the introduction and contributions should be presented in bullet points. 
